# Two-Step One-Pot Reductive Amination of Furanic Aldehydes Using CuAlO_x_ Catalyst in a Flow Reactor

**DOI:** 10.3390/molecules25204771

**Published:** 2020-10-17

**Authors:** Alexey L. Nuzhdin, Marina V. Bukhtiyarova, Valerii I. Bukhtiyarov

**Affiliations:** Boreskov Institute of Catalysis SB RAS, 630090 Novosibirsk, Russia; mvb@catalysis.ru (M.V.B.); vib@catalysis.ru (V.I.B.)

**Keywords:** reductive amination, 5-hydroxymethylfurfural, 5-acetoxymethylfurfural, Cu-based catalyst, Cu-Al mixed oxide, primary amines, imine hydrogenation, flow reactor

## Abstract

Aminomethylhydroxymethylfuran derivatives are well known compounds which are used in the pharmaceutical industry. Reductive amination of 5-hydroxymethylfurfural (HMF) derived from available non-edible lignocellulosic biomass is an attractive method for the synthesis of this class of compounds. In the present study, the synthesis of N-substituted 5-(hydroxymethyl)-2-furfuryl amines and 5-(acetoxymethyl)-2-furfuryl amines was performed by two-step process, which includes the condensation of furanic aldehydes (HMF and 5-acetoxymethylfurfural) with primary amines in methanol on the first step and the reduction of obtained imines with hydrogen in a flow reactor over CuAlO_x_ catalyst derived from layered double hydroxide on the second step. This process does not require isolation and purification of intermediate imines and can be used to synthesize a number of aminomethylhydroxymethylfurans in good to excellent yield.

## 1. Introduction

N-Substituted 5-(hydroxymethyl)-2-furfuryl amines are an important class of compounds due to their usage in pharmaceutical industry for production of calcium antagonists, muscarinic agonists, cholinergic agents, carcinogenesis inhibitors, etc. (for instance, anti-hepatitis B virus drug and antihypertensives, Scheme 1) [1,2,3]. Generally, these structures are prepared by a Mannich-type reaction starting from furfural alcohol (or furfural), formaldehyde, and primary amines. However, harsh reaction conditions are usually required, that leads to a low yield of the target product [4,5,6]. Meanwhile, reductive amination of 5-hydroxymethylfurfural (HMF) derived from available non-edible lignocellulosic biomass is an attractive method for the synthesis of this type of amines [3,7,8,9,10,11].

In general, the reductive amination of HMF is realized in the presence of noble metal-based catalysts such as dichlorobis(2,9-dimethyl-1,10-phenanthroline) ruthenium(II) complex [7], Pd/C [8], Au/TiO_2_ [3], and Pd nanoparticles immobilized in a MOF/polymer composite [9] using batch reactors. However, the high price and limited availability of precious metals are stimulated interest in catalysts based on earth-abundant elements. Chieffi et al. studied the reductive amination of HMF over carbon-supported Fe-Ni catalyst under flow conditions. However, the amines used were limited to 3-aminopropanol and alanine sodium and the yield of the target aminomethylhydroxymethylfuran (AMHMF) derivatives did not exceed 78% [11].

In our previous study [12], an environmentally friendly procedure has been developed for the synthesis of secondary amines through a two-step one-pot reductive amination of aromatic aldehydes with primary amines. The condensation of benzaldehyde derivatives (or furfural) with amines in methanol at room temperature gives imines, which are then hydrogenated in a flow reactor using Cu-Al mixed oxide (denoted hereafter as CuAlO_x_) derived from layered double hydroxide (LDH). In present work, we investigated the efficiency of this procedure for reductive amination of furanic aldehydes (HMF and 5-acetoxymethylfurfural) with primary amines via hydrogenation of the intermediate imines over CuAlO_x_ catalyst in a flow reactor.

## 2. Results and Discussion

Methanol is a suitable solvent for the synthesis of N-substituted 5-(hydroxymethyl)-2-furfuryl amines, since it is a good solvent for both the condensation of aromatic aldehydes with primary amines and hydrogenation of imines [10,12,13]. We found that the reaction of HMF with aniline in low water methanol (0.02 wt% H_2_O) occurs at room temperature with imine yield of 98% (Table 1, entry 1) and time dependence of imine yield let us to see that equilibrium concentrations are reached within 2 h (Appendix A). The use of ethanol instead of methanol leads to a decrease in the yield of imine **1a** to 70% (Table 1, entry 2). Simultaneously, the yield of **1a** in isopropanol is only 31% under the same reaction conditions (Table 1, entry 3).

The reaction mixtures obtained by condensation of HMF with primary amines were further hydrogenated over CuAlO_x_ catalyst in a flow reactor at 100 °C and H_2_ pressure of 10 bar. Before each catalytic run, the CuO phase of CuAlO_x_ was reduced in situ by hydrogen to metallic copper particles [12,14]. The average size of copper crystallites in the reduced sample was about 7 nm [14] that is much smaller than in case of the supported Cu/Al_2_O_3_ catalysts with high copper content. Thus, CuAlO_x_ material prepared by calcination of LDH precursor is a promising non-noble metal catalyst for hydrogenation reactions [12,14,15].

The hydrogenation of imine **1a**, which is obtained by condensation of HMF and aniline in methanol, leads to the formation of AMHMF derivative **2a** with the yield of 97% (Table 2, entry 1). In addition to **2a**, undesired bis-(hydroxymethyl)furan **3** was observed among the reaction products. The use of ethanol or isopropanol instead of methanol leads to a strong drop in the yield of the target product (Table 2, entries 2 and 3) due to the much lower yield of imine in the first stage (Table 1). As a result, a greater amount of HMF is hydrogenated to **3**. Raising the concentration of reactants in methanol to 0.2 M reduces the yield to 94% (Table 2, entry 4). However, an increase of reaction temperature to 120 °C at high reactant concentration leads to rise of the imine hydrogenation rate and allows synthesizing **2a** with the yield of 98% (Table 2, entry 5).

The effect of primary amine structure on the formation of AMHMF derivatives was investigated by reductive amination of HMF with different aromatic and aliphatic amines over CuAlO_x_ catalyst (Table 2, entries 1 and 6–16). Formation of **2b**, **2c**, and **2e** with the excellent yield (95–97%) is observed for aromatic amines containing electron-donating substituent (CH_3_ and OCH_3_) in *meta*- and *para*-position (Table 2, entries 6, 7, and 9). It means that the rates of imine formation in the condensation of HMF with these compounds are similar to that in the reaction of HMF with aniline while using *o*-toluidine gives lower rate of imine formation due to steric hindrance created by methyl group in *ortho*-position [12]. A much longer period of time (16 h) is needed for the condensation of HMF with *o*-toluidine at the first stage to obtain **2d** with the yield of 88% by imine hydrogenation over the CuAlO_x_ catalyst at the second stage (Table 2, entry 8).

Introduction of F and Cl substituent in the *para*-position of aniline slightly decreases the yield of compounds **2** to 93–94% (Table 2, entries 10 and 11). At the same time, in the case of *p*-bromoaniline the significant drop in the yield of the target product to 60% was observed while imine amount in the final mixture was increased (Table 2, entry 12). It is worth to mention that reaction was performed at 110 °C that is higher than in case of other aromatic amines. This result can be explained by the steric hindrance created by bulky Br substituent during the diffusion of imine to active sites of CuAlO_x_ [12]. It should be noted that hydrodehalogenated product practically did not form during the reaction.

The position of electron-withdrawing substituent in aromatic amine has a strong effect on the reaction. The imine formation rate in the condensation of HMF with *m*-chloroaniline is much lower than with *p*-chloroaniline. As a result, the reaction of HMF with *m*-chloroaniline for 3 h on the first stage leads to formation of **2i** in a low yield (Table 2, entry 13). However, carrying out the condensation of HMF and *m*-chloroaniline for 16 h allows obtaining **2i** in 95% yield (Table 2, entry 14). At the same time, *o*-chloroaniline gave a moderate yield of **2j** (52%) after the reaction with HMF on the first stage for 16 h (Table 2, entry 15). These observations are explained by a decrease in the nucleophilic properties of chloroanilines in the following order: *p*-chloroaniline > *m*-chloroaniline > *o*-chloroaniline.

Thus, the yield of **2** is determined by the imine formation at the first stage. Higher yield of the intermediate imine provides higher yield of AMHMF derivative. In the case of primary amines with weak nucleophilic properties (*o*-toluidine, *m*-chloroaniline, *o*-chloroaniline), a longer condensation time (16 h) is required to obtain compound **2** with a higher yield.

Using aliphatic hexylamine gave a slightly lower yield of AMHMF derivative than aniline (Table 2, entries 1 and 16) that can be explained by instability of intermediate imines containing alkyl substituent [16].

The time-on-stream measurements for reductive amination of HMF with aniline over the CuAlO_x_ catalyst were performed for~3 h (Appendix A). The yield of **2a** (97%) remained unchanged meaning that the catalyst exhibits good stability during the reaction [12,14].

5-Acetoxymethylfurfural (AMF), which can be prepared from carbohydrates without isolating HMF, is less reactive, more stable and easily extracted from the aqueous reaction mixture. Therefore, AMF represents a good substitute for HMF [17,18,19]. Similar to HMF, AMF reacts with aniline in methanol to form imine with the yield of 98% (Appendix A). The hydrogenation of imines obtained by condensation of AMF with different amines was carried out on the CuAlO_x_ catalyst at lower temperature of 80 °C than in case of HMF in order to avoid the reduction of the acetoxymethyl group. It was found that reductive amination of AMF with aniline and its derivatives containing CH_3_ group in *para*- and *meta*- positions leads to the formation of N-substituted 5-(acetoxymethyl)-2-furfuryl amines **6a–c** with the yields about 99% (Table 3, entries 1–3). Introduction of methyl substituent in *ortho*-position of aniline decreases the yield of compound **6** to 96% (Table 3, entry 4). It is worth to mention that condensation of AMF with *o*-toluidine was performed for 16 h while temperature of following hydrogenation of obtained imine was raised to 90 °C. In the case of anilines with CH_3_O, F, and Cl substituents in *para*-position, AMHMF derivatives **6e**, **6f**, and **6g** were obtained with the yield of 99, >98 and 97%, respectively (Table 3, entries 5–7). Simultaneously, the reaction of AMF with *m*-chloroaniline for 16 h at the first stage, followed by imine hydrogenation at 80 °C, allows synthesizing **6h** with the yield of 96% (Table 3, entry 8). In contrast to aromatic amines, hexylamine gives only 82% yield of the AMHMF derivative due to hydrogenation of acetyl group in **6i** (Table 3, entry 9).

## 3. Materials and Methods

### 3.1. Chemicals

Aniline (99.8%), *o*-toluidine (99%), *m*-toluidine (99%), *p*-toluidine (99%), *p*-anisidine (99%), *o*-chloroaniline (>98%), *m*-chloroaniline (99%), *p*-chloroaniline (98%), *p*-fluoroaniline (98%), *p*-bromoaniline (>99%), *n*-hexylamine (99%), and 5-(acetoxymethyl)-2-furaldehyde (97%) from Acros Organics (Geel, Belgium), as well as 5-hydroxymethylfurfural (99%) from Sigma-Aldrich (St Louis, MO, USA), were used without additional purification. Methanol (99.8%) from J.T. Baker (Phillipsburg, NJ, USA), ethanol (99.9%) from J.T. Baker and isopropanol (99.5%) from Acros Organics were employed as a solvent.

### 3.2. Catalyst Preparation

Cu-Al layered double hydroxide was prepared by co-precipitation method at pH 9.0 and temperature of 70 °C using the mixture of copper (II) nitrate and aluminum nitrate as starting material and solution containing NaOH and Na_2_CO_3_ ([CO_3_^2^^−^]/[Al^3+^] = 0.86, [OH^−^] = 1.6([Al^3+^] + [Cu^2+^]) as precipitant agent. The aging of the obtained suspension was performed at 70 °C for 4 h. Afterwards, the precipitate was filtered, washed with hot water and dried at 110 °C for 14 h. The as-prepared LDH was calcined at 650 °C for 4 h to obtain desired CuAlO_x_ catalyst [11,13] containing 47.2 wt% of Cu and 19.9 wt% of Al. The texture properties of CuAlO_x_ were found to be as follows: BET specific surface area—68 m^2^ g^−1^ and total pore volume—0.15 cm^3^ g^−1^.

### 3.3. Characterization Techniques

The Cu and Al content was measured by atomic absorption spectroscopy on an Optima 4300 DV instrument (Perkin Elmer, Waltham, MA, USA). Textural characteristics were determined from nitrogen adsorption–desorption isotherms (77 K) obtained on a Micromeritics ASAP 2400 analyzer (Micromeritics, Norcross, GA, USA).

### 3.4. General Procedure for Reductive Amination of Furanic Aldehydes with Primary Amines

Depending on the nucleophilic properties of the amine, the solution of furanic aldehyde (0.05 M) and primary amine (0.05 M) in methanol was kept at 25 °C for 3 or 16 h. Then, the reaction mixture was mixed with H_2_ and pumped through the flow reactor packed with the CuAlO_x_ material. Catalytic experiments were performed in H-Cube Pro setup (Thalesnano, Budapest, Hungary) equipped with a 30 mm CatCart cartridge (CatCart^®^30, 0.30 mL empty volume) [11,13]. Before the catalytic run, the catalyst (0.165 g of 250–500 μm particles) was reduced by a mixture of hydrogen with methanol at 120 °C for 1 h (pressure of 10 bar, flow rates of methanol and H_2_ were 0.5 and 30 mL min^−1^, respectively). Afterwards, the reaction mixture was pumped through the reactor instead of pure solvent, and this point in time was chosen as the starting point of the experiment.

The reaction was carried out at temperature of 80–120 °C, H_2_ pressure of 10 bar, liquid feed rate of 0.5 mL min^−1^ and hydrogen flow rate of 30 mL min^−1^ (inlet H_2_/substrate molar ratio was 54). The use of a large excess of H_2_ in the reaction medium is necessary to avoid the influence of external mass transfer on the reaction progress [13]. The performance of the catalyst was evaluated by analysis of the samples taken in the interval of 30–33 min from the beginning of the experiment. The composition of the reaction products was determined using ^1^H NMR spectroscopy in CDCl_3_. The error in determining the yield is ±1%.

## 4. Conclusions

The two-step one-pot reductive amination of furanic aldehydes, including non-catalytic condensation of HMF (or AMF) with primary amines and the subsequent flow hydrogenation of the obtained imine over Cu-based catalyst derived from layered double hydroxide, was successfully performed using methanol as a solvent. This inexpensive and environmentally friendly process can be utilized to obtain a wide range of N-substituted 5-(hydroxymethyl)-2-furfuryl amines and 5-(acetoxymethyl)-2-furfuryl amines in good to excellent yields (up to 98 and 99%, respectively). The proposed approach is a new method for the synthesis of aminomethylhydroxymethylfuran derivatives which can be used as pharmaceuticals or their precursors.

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
