# Peer review of "Two-Step One-Pot Reductive Amination of Furanic Aldehydes Using CuAlO_x_ Catalyst in a Flow Reactor"

_molecules, 2020, doi:10.3390/molecules25204771_

Round 1
Reviewer 1 Report
Manuscript Number: Molecules 962404
Title: Two-step one-pot reductive amination of furanic aldehydes using CuAlOx catalyst in a flow reactor
Authors: Alexey L. Nuzhdin, Marina V. Bukhtiyarova and Valerii I. Bukhtiyarov
The authors used a flow reactor with CuAlOx catalyst for the Reductive amination of 5-hydroxymethylfurfural using a two step process. In the first step intermediates imines were produced while in the second step the catalytic amination occurred. The catalyst CuAlOx was derived from layered double hydroxide in the second step. The advantage of this process is that no isolation or purification is needed for the production of the final products.
The work is well designed and easy to follow. The manuscript, although short in length can be accepted for publication after major revision.
Comments
The authors should revised their introduction part and discuss in it their previous results. The novelty of the work should be emphasized
Figure 1 is not informative. It can be deleted as the authors discuss the stability of the process in the text
The conversion of HMF and AMF in the first step can be added in text (if different from 100%). Also, if the authors have performed kinetics experiments for the first stage the provided results can improve significant the manuscript.
Can the authors explain why some reactants demand higher time (16 h) for the conversion?
What is the meaning of “reaction time 30–33 min” in tables 1 and 2, for a fixed bed reactor? Please explain or edit in the manuscript
The 3.5 h experiment is not so long to reveal inactivation phenomena. Higher times required.
Do the authors characterized their catalyst after reduction with H2? Their conditions are capable for the full reduction of Cu phase?
Author Response
>Comments
>The authors should revised their introduction part and discuss in it their previous >results. The novelty of the work should be emphasized
Response. Agree with Referee. The introduction part is extended by a discussion of the results reported in [1].
[1] J. Chem. Technol. Biotechnol., 2020, DOI: 10.1002/jctb.6508.
>Figure 1 is not informative. It can be deleted as the authors discuss the stability >of the process in the text
Response. Agree with Referee. Figure 1 is moved to Supplementary Materials.
>The conversion of HMF and AMF in the first step can be added in text (if different >from 100%). Also, if the authors have performed kinetics experiments for the >first stage the provided results can improve significant the manuscript.
Response. Agree with Referee. Table 1 containing the results of the reaction of HMF with aniline in various alcohols at room temperature is added to the manuscript. In addition, the kinetic profiles for condensation of HMF (or AMF) with aniline are added to Supplementary Materials.
>Can the authors explain why some reactants demand higher time (16 h) for the >conversion?
Response. In the case of primary amines with weak nucleophilic properties (o-toluidine, m-chloroaniline, o-chloroaniline), a longer condensation time (16 h) is required to obtain compound 2 with a higher yield. This sentence is added to the manuscript.
>What is the meaning of “reaction time 30–33 min” in tables 1 and 2, for a fixed >bed reactor? Please explain or edit in the manuscript
Response. Before the catalytic run, the catalyst was reduced by a mixture of hydrogen with methanol at 120 °C for 1 h. Afterwards, the reaction mixture was pumped through the reactor instead of pure solvent, and this point in time was chosen as the starting point of the experiment. The performance of the catalyst was evaluated by analysis of the samples taken in the interval of 30–33 minutes from the beginning of the experiment.
On the recommendation of the reviewer, the phrase “reaction time 30–33 min” is removed from the footnotes of Tables 2 and 3.
>The 3.5 h experiment is not so long to reveal inactivation phenomena. Higher >times required.
Response. We agree that approximately 3 h of experiment is not enough to study catalyst deactivation. However, it has previously been observed that many heterogeneous metal catalysts rapidly lose their activity during liquid-phase or gas-phase hydrogenation of N-containing organic compounds (for example [1-6]). In the case of СuAlOx catalyst, the time-on-stream test in the reductive amination of HMF with aniline showed that the yield of 2a (97%) remained unchanged for 3 hours. This allows us to conclude that the catalyst exhibits good stability during the reaction.
[1] Catal. Sci. Technol., 2015, 5, 4741 – 4745.
[2] Catal. Commun. 2017, 102, 108-113.
[3] Chem Eng J, 2014, 255, 695–704
[4] Green Chem, 2007, 9, 849–851
[5] Catal Sci Technol, 2013, 3, 454–461
[6] Catal. Lett., 2017, 147, 572–580
>Do the authors characterized their catalyst after reduction with H2? Their >conditions are capable for the full reduction of Cu phase?
Response. In our previous works [1-4] we showed that the CuO phase of Cu-based catalysts (Cu/Al2O3, CuAlOx, Cu/SiO2, Cu/TiO2-SiO2,) can be reduced to metallic copper particles by a mixture of hydrogen with organic solvent (methanol, toluene) at 120 °C for 1 h (flow rates of solvent and H2 are 0.5 and 30 mL min−1, respectively). The supported catalysts reduced in this way were characterized by XRD, which showed full (or almost full) reduction of copper (II) to metallic copper [1,2]. The treatment of the CuAlOx material in hydrogen flow at 120 °C for 1 h promotes the reduction of CuO to Cu0. The XRD pattern of the reduced CuAlOx sample also contains small peaks associated with Cu2O phase, which was probably formed under the influence of atmospheric oxygen during the transfer of the sample from the catalytic reactor to the spectrometer [3]. Thus, we believe that the CuO phase in the CuAlOx material is completely reduced under experimental conditions.
[1] Catal. Commun. 2017, 102, 108-113
[2] Kinetics and Catalysis, 2018, Vol. 59, No. 5, P. 593-600
[3] J. Chem. Technol. Biotechnol., 2020, DOI: 10.1002/jctb.6508
[4] Molecular Catalysis 494 (2020) 111132
Reviewer 2 Report
The authors report an application of their newly disclosed reductive amination protocol to hydroxymethyl furfural and its acetyl derivative.
In essence, they found that their approach based on a CuAlOx and flow reactor works with good results also for these compounds, which are of high current interest due to their renewable nature. They are in fact considered to be increasingly relevant platform molecules, thus making their efficient manipulation a worthy goal.
On these grounds, I recommend publication of this paper in Molecules, with just a minor modification:
-I think the importance of the paper would better stand out if some examples of these important molecules were added in the introduction. Furthermore, I recommend to cite Green Chemistry 2018, 20, 2494.
Author Response
>I think the importance of the paper would better stand out if some examples of >these important molecules were added in the introduction.
Response. Agree with Referee. Examples of bioactive molecules based on N-substituted 5-(hydroxymethyl)-2-furfuryl amines are added to the introduction section.
>Furthermore, I recommend to cite Green Chemistry 2018, 20, 2494.
Response. Agree with Referee. This reference is added.
Reviewer 3 Report
The authors investigated the efficiency of the procedure for the reductive amination of furanic aldehydes (HMF and 5-acetoxymethylfurfural) with primary amines through hydrogenation of the intermediate imines over Cu-Al mixed oxide in a flow reactor.
Although the overall results seem interesting, I suggest calculation of catalytic parameters: conversion, selectivity, TOF, and TON values, besides the yield parameter stated in the tables. The kinetic profile of HFM conversion is already presented in Fig 1 and the calculation of the selected parameters should not be problematic.
Furthermore, have the authors tested the addition of some other solvents besides MeOH? Maybe some other alcohol or other organic solvents?
Due to the above mentioned reasons aI suggest major revision.
Author Response
>Although the overall results seem interesting, I suggest calculation of catalytic >parameters: conversion, selectivity, TOF, and TON values, besides the yield >parameter stated in the tables. The kinetic profile of HFM conversion is already >presented in Fig 1 and the calculation of the selected parameters should not be >problematic.
Response. Kinetic profiles for condensation of HFM with aniline at room temperature in various alcohols are added to Supplementary Materials. However, the study of the kinetics of imine hydrogenation at the second stage is a difficult task because the hydrogenation products (compounds 2) cannot be analyzed by gas chromatography (decomposition of aminomethylhydroxymethylfuran derivatives occurs during the evaporation of the sample). Moreover, the analysis of the final reaction mixtures containing a large amount of imine by 1H NMR is also difficult due to the possible hydrolysis of the imine during the removal of methanol from the sample.
>Furthermore, have the authors tested the addition of some other solvents >besides MeOH? Maybe some other alcohol or other organic solvents?
Response. The experiments in ethanol and isopropanol are added to the manuscript. The use of ethanol or isopropanol instead of methanol leads to a severe drop in the yield of the target product (2a) due to the much lower yield of imine in the first stage (condensation of HMF and aniline). As a result, a greater amount of HMF is hydrogenated to bis-(hydroxymethyl)furan 3.
Round 2
Reviewer 1 Report
The authors revised their manuscript according to my suggestions. The manuscript can be accepted for publication
Reviewer 3 Report
Since the authors provided all the suggestions and corrections I agree to accept this manuscript for the publication.